



# Using OCO-2 column CO₂ retrievals to rapidly detect and estimate biospheric surface carbon flux anomalies

Andrew F. Feldman[1,2], Zhen Zhang[3], Yasuko Yoshida[4], Abhishek Chatterjee[5], Benjamin Poulter[1]

5 [1]Biospheric Sciences Laboratory, NASA Goddard Space Flight Center, Greenbelt, MD, 20771, USA
[2]NASA Postdoctoral Program, NASA Goddard Space Flight Center, Greenbelt, MD, 20771, USA
[3]Earth System Science Interdisciplinary Center, University of Maryland, College Park, MD, 20740, USA
[4]Science Systems and Applications, Inc. (SSAI), Lanham, MD, 20706, USA
[5]Jet Propulsion Laboratory, California Institute of Technology, Pasadena, CA, 91109, USA

*Correspondence to*: Andrew F. Feldman (afeld24@mit.edu)

**Abstract.** The global carbon cycle is experiencing continued perturbations via increases in atmospheric carbon concentrations. Greenhouse gas satellites that are designed to retrieve atmospheric carbon concentrations can help observe seasonal to interannual variations and sources of carbon dioxide. Recent work has shown that these satellites are further applicable beyond 15 their design specifications for directly identifying and quantifying surface emissions at various spatiotemporal scales. For example, simple mass balance approaches have been used to rapidly estimate surface methane fluxes from satellite atmospheric column methane retrievals. However, less attention has been placed on using satellite column CO₂ retrievals to evaluate surface CO₂ fluxes from the terrestrial biosphere at shorter timescales without inversion models. Such applications could be useful to monitor, in near-real time, biosphere carbon fluxes during climatic anomalies like drought, heatwaves, and floods, before more 20 complex terrestrial biosphere model outputs become available. Here, we explore the ability of Orbiting Carbon Observatory-2 (OCO-2) column-averaged dry air CO₂ (XCO₂) retrievals to directly detect and estimate terrestrial biosphere CO₂ flux anomalies using a simple mass balance approach. Using CarbonTracker model reanalysis as a testbed, we first demonstrate that a previously developed, mass balance approach can estimate monthly surface CO₂ flux anomalies from XCO₂ enhancements in the Western United States. The method is optimal when the chosen target region is spatially extensive enough 25 to account for atmospheric mixing and has favorable advection conditions with contributions primarily from one background region. While errors in individual soundings partially reduce the ability of OCO-2 XCO₂ to estimate more frequent, smaller surface CO₂ flux anomalies, we find that OCO-2 XCO₂ can often detect and estimate larger surface flux anomalies. OCO-2 is thus useful for near real time monitoring of the monthly timing and magnitude of regional terrestrial biosphere carbon anomalies. Any noise reduction in forthcoming greenhouse gas satellites and/or the existence of large surface carbon anomalies 30 will likely enhance the ability to rapidly estimate surface fluxes at smaller spatiotemporal scales.



# 1 Introduction

With ongoing anthropogenic emissions, atmospheric carbon concentrations continue to rise and alter the global climate system (Friedlingstein et al., 2022). Given many diverse carbon sources and sinks as well as transport of atmospheric carbon, it is a
challenge to monitor these atmospheric carbon concentrations and carbon fluxes across the globe. Carbon measurement networks are available, but are spatially biased toward mid-latitude locations with little coverage in the tropics (Schimel et al., 2015a). Therefore, atmospheric transport model assimilation efforts and land surface models are often used to quantify and monitor global carbon sources and sinks (Ott et al., 2015; Peters et al., 2007). However, these datasets typically have a longer latency and complex sources of error due to modeling assumptions about uncertain surface carbon flux drivers.


Greenhouse gas satellites provide an ability to retrieve atmospheric column carbon concentrations and constrain global carbon cycle processes. Over the past two decades, satellite instruments such as SCanning Imaging Absorption spectroMeter for Atmospheric CartograpHY (SCIAMACHY), Greenhouse Gases Observing Satellite (GOSAT), and Orbiting Carbon Observatory-2 (OCO-2) have provided measurements of dry-air column carbon dioxide ($XCO_2$) and in many cases also
methane ($XCH_4$) (Bovensmann et al., 1999; Eldering et al., 2017b; Kuze et al., 2014; Reuter et al., 2011). These satellites have indeed provided observational constraints on carbon-cycle seasonal and interannual variability with global coverage as desired beyond irregular spatial coverage of in-situ networks (Chen et al., 2021; Lindqvist et al., 2015). Additionally, since these column retrievals are partly a function of surface carbon fluxes (Keppel-Aleks et al., 2012), previous studies have assimilated these $XCO_2$ and $XCH_4$ retrievals into atmospheric inversion model frameworks to improve surface carbon flux estimates (Basu
et al., 2013; Chevallier et al., 2014; Fraser et al., 2014; Halder et al., 2021; Houweling et al., 2015; Liu et al., 2017; Ott et al., 2015; Zabel et al., 2014).

It has not been widely investigated whether satellites like OCO-2 can directly monitor the timing and magnitude of shorter monthly timescale climate-carbon feedback events, such as those that evolve in the terrestrial biosphere and generate regional
and short-lived $XCO_2$ enhancements. OCO-2 was designed to observe regional-scale carbon sources and sinks to provide a constraint on carbon cycle seasonal and interannual variability (Crisp et al., 2004; Eldering et al., 2017b). Despite initial concern that the noise level of individual soundings would prevent direct monitoring of surface $CO_2$ flux evolution at finer scales (Chevallier et al., 2007; Eldering et al., 2017a; Miller et al., 2007), there is growing evidence that satellite $XCO_2$ retrievals can directly detect and monitor surface carbon sources on smaller spatiotemporal scales. For example, many studies
have demonstrated that OCO-2 and other satellites can detect anthropogenic emission plumes from urban areas using spatially adjacent satellite soundings (Hakkarainen et al., 2016; Irakulis-Loitxate et al., 2021; Kort et al., 2014; Nassar et al., 2017; Reuter et al., 2019; Schwandner et al., 2017; Zheng et al., 2020). For natural emission sources, these satellites are typically used to evaluate effects of an event averaged over seasons or multiple years, such as El Niño Southern Oscillation events and related biomass burning (Byrne et al., 2021; Chatterjee et al., 2017; Eldering et al., 2017b; Hakkarainen et al., 2019; Heymann





et al., 2016; Liu et al., 2018; Patra et al., 2017). However, recent studies interpret monthly $XCO_2$ anomalies (from OCO-2) without inversion models to understand evolution of a flood event (Yin et al., 2020) and ocean and terrestrial biosphere carbon cycle co-evolution during the 2015-2016 El Niño (Chatterjee et al., 2017). As such, satellite $XCO_2$ shows promise for directly monitoring the monthly timing and evolution of regional carbon-climate feedbacks from the biosphere at smaller spatiotemporal scales without model assimilation frameworks. Directly observing surface fluxes with satellite $XCO_2$ would

allow rapid detection, monitoring, and/or estimation of surface $CO_2$ fluxes, which are only sparsely observed with ground networks and become available at longer latency from modeling efforts (Schimel et al., 2015a)

A main consideration of evaluating $XCO_2$ over smaller spatiotemporal scales is whether the surface carbon perturbation to $XCO_2$ is above the $XCO_2$ retrieval noise level. Surface carbon sources and sink anomalies ultimately need to be detected within

a high background $XCO_2$ variability, driven by atmospheric transport as well as surface fluxes (Basu et al., 2018; Hakkarainen et al., 2016). The observed $XCO_2$ anomalies are typically under 2 ppm (Hakkarainen et al., 2019). Furthermore, $XCO_2$ anomalies attributed to the most extreme surface perturbations may be below 1 ppm (Chatterjee et al., 2017; Crisp et al., 2017; Miller et al., 2007; Weir et al., 2021). However, GOSAT and SCIAMACHY $XCO_2$ retrievals have estimated uncertainty over 1 ppm (Buchwitz et al., 2017a; Butz et al., 2011), which limits their ability to interpret even the strongest monthly $XCO_2$

anomalies due to surface fluxes. By contrast, OCO-2 $XCO_2$ uncertainty is between 0.5 and 1 ppm for a given sounding (Eldering et al., 2017b; Wunch et al., 2017). With its greater number of daily soundings and higher resolution, aggregated regional-scale $XCO_2$ uncertainty can decrease below 0.5 ppm (Chatterjee et al., 2017). This precision has allowed detection of regional declines in fossil fuel emissions on the order of 0.25-0.5 ppm during the COVID-19 pandemic (Weir et al., 2021), with caveats of limited anomaly detection on the lower end of this range (Buchwitz et al., 2021; Chevallier et al., 2020). As

such, OCO-2 may provide an ability to monitor the evolution of smaller regional surface sources and sinks of $CO_2$ more precisely at monthly timescales than previous spaceborne greenhouse gas instruments.

Recently, $XCH_4$ retrievals have been used to rapidly estimate surface methane fluxes using simple mass balance approaches (Buchwitz et al., 2017b; Pandey et al., 2021). These methods do not require transport models and perform reasonably well.

However, it is unclear whether such an approach can be used for surface fluxes – $CO_2$ fluxes tend to have more spatially homogenous surface sources and sinks compared to more spatially heterogeneous $CH_4$ fluxes. Nearly all efforts to estimate surface $CO_2$ fluxes from OCO-2 $XCO_2$ retrievals have involved transport models and inversions (Byrne et al., 2021; Liu et al., 2017; Palmer et al., 2019; Patra et al., 2017). The few studies estimating surface emissions directly from the $XCO_2$ anomalies alone are empirical (rather than physically-based mass balance methods) in using model-based relationships between $XCO_2$

and surface $CO_2$ fluxes (Heymann et al., 2016) or are specific to point source plumes at under kilometer scales rather than hundreds of kilometer scale areal sources (Zheng et al., 2020).



An equivalent approach using $XCO_2$ in a mass balance would provide an ability to rapidly estimate regional total carbon flux anomalies from the terrestrial biosphere, which are difficult to estimate given their many contributions and require

sophisticated models. Specifically, such a method could allow near real time monitoring of the duration, magnitude, and spatial extent of $CO_2$ flux anomalies during extreme climatic events (Frank et al., 2015; Reichstein et al., 2013). Such applications are especially important for regional climate change hotspots like in the southwestern North America where droughts and heatwaves are becoming more frequent and intense (Cook et al., 2015; Schwalm et al., 2012; Williams et al., 2022). Analogously, a simple approach for estimating ecosystem water fluxes (i.e., triangle method; Carlson, 2007) has a legacy of

continued use given its relatively sufficient accuracy for many applications compared to more complex land surface model approaches. Given the ongoing challenges of estimating surface fluxes at large spatial scales, we anticipate that it will be similarly useful to develop simple total surface carbon flux estimation approaches that are rapid, rely on observations alone (from remote sensing), do not require many modeling assumptions and ancillary data, and provide an independent estimate to evaluate model outputs.


Here, we ask: can satellite retrieved $XCO_2$ be used with mass-balance approaches to directly detect and estimate terrestrial surface $CO_2$ flux anomalies, especially from the biosphere? Can surface $CO_2$ flux anomalies be monitored with $XCO_2$ at sub-seasonal (i.e., monthly) scales? Which wind and spatial domain conditions are most suitable for coupling between $XCO_2$ and surface $CO_2$ fluxes such that $XCO_2$ can be used to estimate surface $CO_2$ fluxes? OCO-2 is chosen due mainly to its high

precision and greater sensitivity to the lower atmosphere, which makes it more sensitive to surface fluxes than other greenhouse gas satellites (Eldering et al., 2017a). Recent algorithmic updates have also been shown to increase OCO-2 $XCO_2$ retrievals' representation of biospheric fluxes at subcontinental scales (Miller and Michalak, 2020).

Addressing these questions here can help assess whether greenhouse-gas satellites can be used to monitor biosphere carbon

responses to climatic anomalies at sub-seasonal timescales and in near real time (within 1–2-month latency). For example, greenhouse gas satellite $XCO_2$ anomalies could be used as an initial assessment of an ongoing extreme event and guide more holistic monitoring and attribution of the event with other observational and model tools. This would provide a rapid carbon cycle monitoring capability not available with global models and sparse networks that have a lack of spatial coverage and/or longer latency (Ciais et al., 2014). These questions will also assess our ability to rapidly estimate regional biosphere fluxes in

climate change hotspots as well as in the tropics (Byrne et al., 2017) which sequester the most fossil fuels but lack measurement networks (Liu et al., 2017; Schimel et al., 2015b).

## 2 Methodology



## 2.1 Region Selection

We evaluate the Western US (latitude of 33° N to 49° N and longitude of 124° W to 104° W) given its extent of natural ecosystems that serve as a carbon sink as well as the fact that it has become a hotspot for droughts, including an ongoing decadal-scale megadrought (Cook et al., 2015; Schwalm et al., 2012; Williams et al., 2022). In this region, we expect that terrestrial biosphere fluxes (i.e., photosynthesis, respiration, wildfire) will dominate $CO_2$ surface flux anomalies rather than anthropogenic fluxes. We later show that this region additionally has favorable advection conditions for using $XCO_2$ to assess
surface fluxes (see Sect. 3.1).

## 2.2 Datasets

The study includes three components to assess the potential for using $XCO_2$ to directly evaluate monthly surface flux anomalies. We first evaluated advection conditions where we use the modern-era retrospective analysis for research and applications, version 2 (MERRA2) wind vectors between the surface and 700 mb, which approximately captures the boundary
layer (Gelaro et al., 2017; GMAO, 2015). In this study, we refer to advection as the horizontal transport of air, especially that in the boundary layer. This lower troposphere layer directly interacts with the surface fluxes that influence $XCO_2$ (Buchwitz et al., 2017b; Pandey et al., 2021).

Second, we tested the ability of $XCO_2$ to estimate surface $CO_2$ fluxes using CarbonTracker model reanalysis (CT2019B) as a
testbed, which assimilates tower eddy flux and satellite atmospheric radiance observations into an atmospheric transport model and outputs hourly $XCO_2$ and total surface $CO_2$ fluxes from 2000 to 2018 (Peters et al., 2007). Tests performed using this model reanalysis dataset are meant to represent simulated "true" relationships between surface fluxes and $XCO_2$ dynamics. However, we acknowledge model errors in this framework. A purely simulated environment with error free conditions is not possible here because coupling between surface $CO_2$ fluxes and $XCO_2$ require modeling and assumptions about atmospheric
transport and emission physics. Therefore, we recognize that error in estimating surface $CO_2$ fluxes from $XCO_2$ will be partially a function of errors in modeling assumptions beyond that of errors in the simple mass balance approach.

Third, we assessed the ability of observed $XCO_2$ between September 2014 and July 2021 from the Orbiting Carbon Observatory 2 (OCO-2) to detect and estimate surface fluxes using mass balance (Eldering et al., 2017b). OCO-2 has an
approximate 3 $km^2$ resolution per sounding and 16-day revisit cycle with soundings at around 1:30 pm local time. We use OCO-2 level 2, bias-corrected, retrospective reprocessing version 10 of $XCO_2$ (OCO-2-Science-Team et al., 2020). Quality flags were used to remove soundings with poor retrievals. Observations of total surface $CO_2$ fluxes are only sparsely located in space. We, therefore, independently estimated surface fluxes from a biosphere model, fire reanalysis, and anthropogenic emission repositories. The Lund-Potsdam-Jena (LPJ) dynamic global vegetation model was driven with MERRA2 reanalysis
forcing to output $CO_2$ flux from net biome production (NBP) between January 1980 and July 2021 (Gelaro et al., 2017; Sitch



et al., 2003; Zhang et al., 2018). NBP models carbon fluxes from photosynthesis, respiration, land use change, and fire. Since LPJ only evaluates fire dynamics at the annual scale and wildfire can rapidly evolve over widespread areas in the Western US, fire carbon fluxes were obtained from Quick Fire Emissions Dataset (QFED) biomass burning emissions between 2000 and 2021 to account for monthly fire dynamics in the total carbon fluxes (Koster et al., 2015). LPJ NBP annual fire emissions were
removed from the total NBP and monthly QFED biomass burning emissions were added. Anthropogenic $CO_2$ fluxes were obtained from CarbonMonitor for the Western US region between 2019 and 2021 (Liu et al., 2020). MODIS-based FluxSat gross primary production (GPP), though only evaluating photosynthesis and no respiration or disturbance components, provides another independent observation-based surface flux estimate to determine coupling between $XCO_2$ and biospheric fluxes (Joiner and Yoshida, 2021, 2020).

**2.3 Wind Vector Analysis**

We assessed monthly averaged MERRA2 wind vectors between the surface and 700 mb across the Western US. We specifically used the eastward and northward wind fields from the MERRA2 assimilated meteorological fields (M2T3NVASM) (Gelaro et al., 2017). We first visually evaluated wind quiver plots. Next, the spatially averaged wind direction and speed were determined within the region and at each of its four borders. The average (or, here, "total") wind
velocity is determined within the region by averaging the velocity from all pixels in the region. The speed of winds entering the region at each border were computed by determining only the average velocity component of the wind entering the region. For example, at the western border, the eastward wind velocity component of the pixels along the western border of the domain were averaged. Finally, the percentage of the background region's boundary layer air entering the domain for each of its four borders was estimated as the ratio of the wind vector component entering the region to the total wind vector. For example, at
the western border of the domain, this percentage is computed as the speed of the eastward component of the wind velocity divided by the total velocity at the border. Negative values are set to zero to indicate that air from that background region does not enter the target domain on average.

**2.4 $XCO_2$-Based Surface Flux Estimation**

First, $XCO_2$ and $CO_2$ surface fluxes in all cases were averaged to monthly and spatially averaged within the Western US target
region. Monthly $XCO_2$ and $CO_2$ surface fluxes were deseasonalized by averaging all months in the available time series into an average 12-month climatology (i.e., all January values were averaged, all February values were averaged, and so on). This average climatology was subtracted from the raw time series to create an anomaly time series. Given that $XCO_2$ includes a strong annual increasing trend, each of the twelve months were individually, linearly detrended first before deseasonalizing as in Chatterjee et al. (2017).




Total surface $CO_2$ flux anomalies were estimated from $XCO_2$ anomalies in the Western US using a simple mass balance approach previously applied to methane fluxes (Buchwitz et al., 2017b; Jacob et al., 2016; Varon et al., 2018). It was similarly used here to estimate surface $CO_2$ fluxes (Q; in units of TgC/mo) using:

$$Q = (\Delta XCO_2)(V)(L)(C)(M_{exp})(M) \qquad (1)$$

$\Delta XCO_2$ (ppm) is the difference in $XCO_2$ between the target domain (here, the Western US) and the background region where the majority of the incoming winds originate. V is the ventilation wind velocity (in m/s units, but converted to km/month), which has been motivated previously to be best represented by boundary layer winds (Buchwitz et al., 2017b; Pandey et al., 2021). Thus, while the full column $CO_2$ concentrations were evaluated, the wind speeds in the lower atmosphere are considered in the mass balance model given their greater degree of interaction with the $CO_2$ fluxes at the surface. Here, V is represented as the monthly averaged boundary layer wind speed within the target region. For the observation-based analysis, MERRA2 wind fields are used while for the reanalysis framework study, CarbonTracker wind velocity outputs are used. L is the effective region length (km) meant to estimate the horizontal pathlength of the ventilation wind passing through the region and interacting with the surface flux. L can be estimated by the square root of the target region area. The model parameter, C, represents a model mass balance assumption that $CO_2$ fluxes are spatially homogenous, and the ventilation wind is uniform and consistent across the region. This means there is a consistent, linear increase of $XCO_2$ spatially (from east to west in the Western US region) as the horizontal winds move over a surface $CO_2$ efflux. C is equal to 2 (unitless) under this assumption where the difference between $XCO_2$ at the entrance and exit of the region (from the perspective of the horizontal wind) is twice that of the spatially averaged $\Delta XCO_2$. $M_{exp}$ (unitless) is an adjustment of the surface pressure, or the ratio of the target region's surface pressure to standard atmospheric pressure. For the observation-based analysis, MERRA2 surface pressure is used while in the analysis using the reanalysis testbed, CarbonTracker surface pressure outputs are used. M converts the atmospheric carbon dioxide mixing ratio (or its concentration) to a total column mass considering the volume of the atmospheric column overlying the target region's land surface. It additionally includes a conversion from $CO_2$ to its carbon equivalent. M is therefore $4.2 \times 10^{-6}$ TgC/(km$^2$ ppmCO$_2$).

Previous demonstrations of Eq. 1 on methane fluxes evaluated the raw, rather than anomaly, $XCH_4$ enhancements (Buchwitz et al., 2017b; Pandey et al., 2021). In the main analysis, we have removed $XCO_2$ seasonality here due to many sources of seasonal atmospheric $CO_2$ variability (atmospheric and surface-based) that contribute to $XCO_2$, which hinders causal attribution of $XCO_2$ changes to surface anomalies. Monthly anomalies of $XCO_2$ can thus be more directly attributed to surface carbon flux anomalies than their raw variations can. However, we also evaluate raw $XCO_2$ enhancements for comparison.

The $\Delta XCO_2$ estimated surface fluxes from Eq. 1 were compared with independently determined surface fluxes using the mean bias, root mean square difference (RMSD), and Pearson's correlation coefficient. Two comparisons were performed: one in a reanalysis framework and another testing OCO-2 observations. In the CarbonTracker reanalysis tests, the surface fluxes were





estimated from CarbonTracker-output $XCO_2$, wind velocity, and surface pressure using Eq. 1 and were compared against
CarbonTracker-output surface $CO_2$ fluxes, which represent the total surface $CO_2$ flux from both natural and anthropogenic
sources. In the observational assessment, OCO-2 $XCO_2$ is used to estimate surface $CO_2$ fluxes along with MERRA2 boundary
layer wind velocity and surface pressure in Eq. 1. These $XCO_2$-based surface $CO_2$ flux estimates are compared to total surface
$CO_2$ fluxes, which were estimated using the sum of LPJ NBP model anomalies, QFED biomass burning anomalies, and
CarbonMonitor fossil fuel estimates. Though CarbonMonitor is only available over a short record, we used the record to
determine that the proportion of anthropogenic flux anomalies in the Western US contribute less than 5% to the surface
anomalies (see Fig. S1). Therefore, we define total flux estimates of $CO_2$ in the observation-based assessment to be the sum
of LPJ NBP and QFED biomass burning anomalies, acknowledging that there may be additional smaller deviations due to
fossil fuel emissions. By varying the target domain region size, these statistics were used to determine the optimal target region
size in the Western US, which is sensitive to errors due to atmospheric mixing. This test ultimately determines whether
atmospheric mixing over monthly timescales confounds the use of Eq. 1. These statistics were additionally used to assess the
optimal wind speed and direction conditions for use of Eq. 1. Namely, these statistics were computed conditioning on
ventilation wind speed and direction. Such wind conditions are highly important to assess in such pixel source mass balance
methods (Varon et al., 2018).

$XCO_2$ anomalies and anomaly enhancements were additionally correlated directly with the surface fluxes to determine the
degree of surface and atmospheric $CO_2$ coupling. Specifically, OCO-2 $XCO_2$ are correlated with the biosphere-only LPJ NBP
and FluxSat GPP anomalies to determine how coupled $XCO_2$ anomalies are to biospheric fluxes at monthly timescales.

OCO-2-retrieved $XCO_2$'s ability to detect the largest surface $CO_2$ fluxes in the Western US is evaluated using:

$$Detection\ Rate_{x,y}\ =\ \frac{N_{\Delta XCO_2>yth\ \&\ Q>xth}+N_{\Delta XCO_2<(1-yth)\ \&\ Q<(1-xth)}}{N_{\Delta XCO_2>yth}+N_{\Delta XCO_2<(1-yth)}}*100 \qquad (2)$$

The detection rate is the percentage of months $XCO_2$ anomaly enhancements ($\Delta XCO_2$) of a specified magnitude detect surface
$CO_2$ fluxes (Q) of a given magnitude. x is the percentile referring to monthly surface $CO_2$ flux anomalies and y is the percentile
referring to corresponding monthly $XCO_2$ anomaly enhancements. N is a count of number of Western US domain-averaged
pairs that satisfy the conditions in Eq. 2. For example, if x is 75 and y is 95, then the detection rate determines the number of
$XCO_2$ anomaly enhancements above the 95th percentile that are coincident with a surface $CO_2$ flux anomaly of 75th percentile
or above. Eq. 2 also considers the carbon uptake cases of negative $XCO_2$ anomaly enhancements below the 5th percentile that
coincide with a negative surface $CO_2$ flux anomaly of 25th percentile or less. We use a convention of positive surface $CO_2$
fluxes being away from the ground. As a property of the detection rate metric, when conditioning on only data pairs when
$XCO_2$ anomaly enhancements are above the 95th percentile (y=95), the detection rate will decrease as the x percentiles are
increased. This metric ultimately provides a measure of information that a given $XCO_2$ anomaly enhancement holds about a
corresponding surface $CO_2$ flux anomaly. This detection rate was compared to detection rates by chance, which are equal to



100-x. We also alternatively evaluated whether the largest surface $CO_2$ efflux anomalies influence the $XCO_2$ anomaly signal by estimating the percentage of largest surface efflux anomalies that $XCO_2$ observes a positive anomaly. We evaluate this metric in several global regions using OCO-2 $XCO_2$ anomalies and FluxSat GPP to determine whether $XCO_2$ anomalies can rapidly detect large surface biosphere $CO_2$ fluxes from extreme events.

Performance of these methods using the observation-based method was compared directly to the CarbonTracker simulation performance, with CarbonTracker tests serving as a "potential" or upper bound on performance given expected $XCO_2$ observation error from OCO-2.

## 2.5 $XCO_2$ Surface Flux Error Estimation

Given known limitations of potentially restrictive greenhouse gas satellite measurement and retrieval errors (Buchwitz et al., 2021), we estimated the effect of these $XCO_2$ errors on carbon flux estimates, especially their impact on $\Delta XCO_2$. OCO-2 $XCO_2$ error standard deviation is typically 0.6 ppm for a given observation. However, subtracting two $XCO_2$ errors and spatial averaging in our procedure in Eq. 1 will have competing amplification and reduction effects on the error standard deviation of $\Delta XCO_2$. Therefore, we estimated the spatially averaged $\Delta XCO_2$ error standard deviation. First, we used a bootstrapping approach to assess subtracting a pair of two $XCO_2$ errors drawn randomly from a normal distribution with mean zero and 0.6 ppm standard deviation. Errors are assumed to be normally distributed, which is a general property (i.e., central limit theorem) arising from addition of independent, identically distributed satellite instrument errors. Second, we evaluated the response of the error estimate to averaging approximately twenty values (the typical number of soundings in each month in the target Western US domain), which provides a spatially averaged $XCO_2$ error for both the target and background regions before subtracting the two values. Third, we investigated the role of positive spatial autocorrelation of errors on the overall $XCO_2$ error standard deviation, where autocorrelation can cause competing effects on $XCO_2$ anomaly enhancement errors. Namely, if the errors are positively spatially autocorrelated within a given area, noise reduction from spatial averaging will be partially prevented. However, a spatially autocorrelated relationship between $XCO_2$ errors of the adjacent target and background regions will reduce $\Delta XCO_2$ errors because the $XCO_2$ errors in the two regions would be positively related.

## 3 Results and Discussion

### 3.1 Advection Condition Assessment

Given that we wish to use $XCO_2$ anomalies in the Western US with only a simple source pixel mass balance method and not an atmospheric transport model and/or assimilation framework to monitor surface $CO_2$ fluxes, an understanding of the existing advection conditions in the selected domain is critical. Therefore, we first assessed the boundary layer advection conditions (Fig. 1). Eq. 1 was previously applied to smaller spatiotemporal scales (within a kilometer) to estimate emissions of spatially





heterogeneous natural or urban methane plumes (Pandey et al., 2021). However, $CO_2$ generally has more spatially homogenous surface sources and sinks. Furthermore, we wish to evaluate fluxes from terrestrial ecosystems that exceed tens of kilometers in spatial scales. The need for larger areas and at the monthly aggregated scales thus includes consideration of more
atmospheric mixing of widespread $CO_2$ surface sources and sinks.

As such, it is necessary to understand the advection conditions needed to apply Eq. 1 over large spatial scales, at monthly timescales, and for $CO_2$ that has more spatially distributed surface contributions than $CH_4$. Under these considerations, the most ideal conditions for Eq. 1 appear to be winds that, on average over a month, consistently originate from a single
background region and flow steadily and consistently (without greatly changing directions) through the region throughout the year. These conditions typically hold in the Western US where winds originate from the Pacific Ocean and flow west to east consistently through the domain (Fig. 1). By contrast, the advection conditions may be more complex in a region like the Southeast US which experiences changes in background source of incoming advection from the Midwestern US in the fall and winter to the Gulf of Mexico in the spring and summer. These variable background conditions and inconsistent wind directions
may create large errors when applying Eq. 1.

A more comprehensive evaluation of the boundary layer wind vectors reveals that the Pacific Ocean at the region's west border comprises the main source of advection entering the target region throughout the year (Fig. 2a). Winds along its northern, southern, and eastern borders have little contribution to the region. This suggests that Eq. 1 can be applied more confidently
in assuming only one background region contributes advection to the Western US. More detailed evaluation of the incoming advection from the Pacific Ocean reveals that incoming winds at the western border and throughout the region are continuously of non-negligible magnitude and consistently to the east (Figs. 2b and 2c). The exception is spring and summer months when winds in the Pacific Ocean shift to the south and the speed of eastward winds into and within the region are lower.

Nevertheless, these conditions initially indicate that the Western US region is a candidate region to detect and estimate surface fluxes using satellite $XCO_2$ retrievals. We test the ability to estimate surface $CO_2$ fluxes from $XCO_2$ under these advection conditions using CarbonTracker reanalysis hereafter.





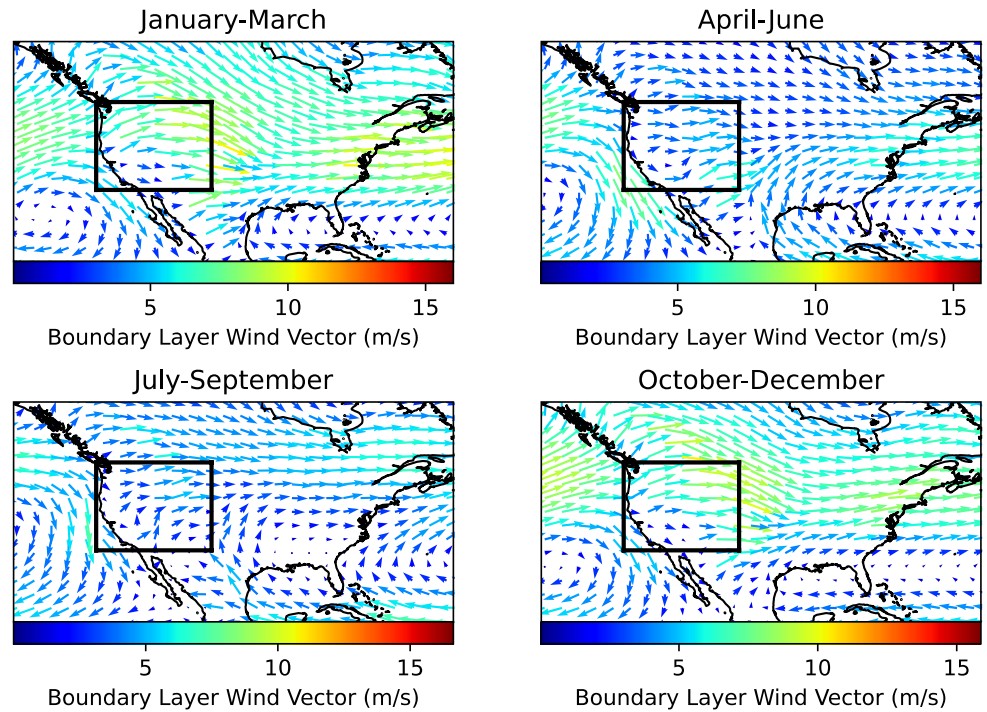

**Figure 1.** Mean monthly boundary layer (surface up to approximately 700 mb) conditions from MERRA2 in each season. The Western US
target domain is identified with borders.

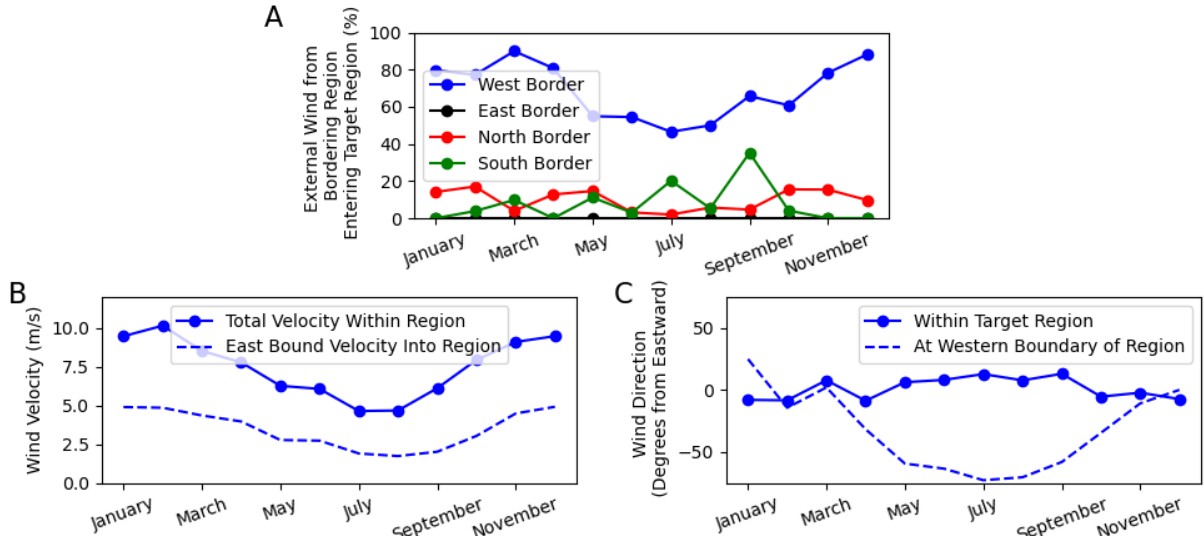

**Figure 2.** Mean monthly advection conditions in the Western US target domain in Fig. 1. (a) Proportion of wind vector entering region from
each bordering region. Values do not add to 100% because each border's wind vector is evaluated individually for its contribution to the





Western US domain. **(b)** Mean wind velocity and **(c)** mean wind direction within the target region and eastward into the target domain from the Pacific Ocean on its western border.

### 3.2 Reanalysis Evaluation

Here, using CarbonTracker (CT2019B) model reanalysis as a testbed, we evaluate by how much the advection conditions present limitations for detecting and estimating surface $CO_2$ fluxes using $XCO_2$ anomalies in the Western US. We tested the
effect of domain area size, wind angle, and wind speed on the mass balance surface flux estimation.

Using CarbonTracker, we find that a smaller target region has a large decline in ability to estimate surface fluxes with Eq. 1 (Fig. 3). This reduction in performance of the simple mass balance model is expected because a smaller area will increase importance of turbulent mixing compared to effects of mean horizontal ventilation wind, as was suggested previously (Varon
et al., 2018). It will also include $XCO_2$ contributions from surface sources outside of the selected target region due to mixing. For example, with turbulent mixing on smaller spatial scales, large $CO_2$ surface effluxes from surfaces adjacent to the region may mix with atmospheric $CO_2$ within the region, causing a larger positive $XCO_2$ anomaly than what can be expected from the surface contributions from within the small target domain itself. In fact, the increasing positive mean bias with smaller selected target areas suggests this may be the case where $XCO_2$ within the target region includes contributions from more
surface area than the mass balance model can predict (Fig. 3c). The decorrelation of the $XCO_2$ surface flux estimates with the modeled surface fluxes with smaller surface areas further supports this claim where external $XCO_2$ anomaly variations may be effectively contributing to the $XCO_2$ anomalies within the domain (Fig. 3a). Ultimately, this target region size analysis motivates choosing larger target areas for application of Eq. 1 on $CO_2$ flux estimation, especially over monthly timescales.

In the context of this application for studying the terrestrial biosphere, we speculate that choosing larger regional domains creates greater requirements on the need for consistent advection conditions over a larger spatial scale, limiting the number of areas of the globe where such a method can be applied. If boundary layer winds tend to change directions within the domain or originate from multiple background regions, a smaller domain may simplify the conditions for use of Eq. 1. However, the choice of a smaller area will have competing effects of an increased error due to mixing and turbulence as suggested by Fig. 3
and previous work (Varon et al., 2018). As such, in cases of more complex advection conditions, regions the size of that in Fig. 3e may be optimal.

While variations in the monthly averaged wind speed through the Western US (typically ranging from 1 m/s to 10 m/s) does not consistently impact $XCO_2$ surface flux estimation, large deviations in wind angle from the eastward direction incoming
from the Pacific Ocean can negatively impact flux estimation (Fig. 4). Namely, there is a general reduction in the mass balance equation's (Eq. 1) ability to estimate surface fluxes with increasing wind angle (Fig. 4a). While absolute errors only weakly linearly increase with wind angle (r = 0.12; p-value=0.07) (RMSD's correlation with wind angle is r = 0.53, p-value = 0.16





with only eight bin samples), a more frequent occurrence of higher errors occurs above absolute angles of 60 degrees from the eastward plane (Fig. 4a). As such, the switch to northerly winds in the Pacific Ocean during the summer months (Fig. 2c) may

the cause of seasonally increased surface $CO_2$ flux estimation errors (Fig. 4c). This is expected because the advection of air from the Pacific Ocean into the Western US would be reduced, creating a disconnect between the atmospheric carbon concentrations of the Western US target region and Pacific Ocean in these months (Fig. 1). The effect of variations in monthly averaged wind speed appears to have less of an influence on errors than wind angle (Fig. 4b). Specifically, there is a correlation of 0.02 (p-value=0.7) between wind speed and absolute errors. However, we expect errors would increase if wind speeds

approach zero. Wind speed may also become a larger error source when investigating shorter time steps or more spatially heterogeneous anthropogenic plumes (Jacob et al., 2016; Varon et al., 2018).

Overall, the simple mass balance estimation of surface fluxes with $XCO_2$ (Eq. 1) appears robust to the range of advection conditions in the Western US, except potentially for summer months when Pacific Ocean winds can shift from westerlies to

northerlies (Fig. 5). $XCO_2$ anomaly enhancements are positively correlated with surface flux anomalies in the Western US, especially in cases when the wind angle has a non-negligible eastward component (Fig. 5a). This correlation is maintained when applying Eq. 1 to estimate surface fluxes (Fig. 5b). The comparison improves when consideration of winds that have a wind angle from the eastward reference of less than 60 degrees, or between -60 and 60 degrees when 0 degrees is to the east (Fig. 5). However, we caution that an RMSD of ~20 TgC per month suggests that the approach should be used only as a rapid,

first estimation of surface $CO_2$ flux anomalies.

We additionally show that the method can estimate the surface $CO_2$ fluxes using the raw $XCO_2$ enhancements (i.e., not anomalies), especially when winds have a substantial eastward component (Fig. 5c). However, using the $XCO_2$ anomalies removes seasonal $XCO_2$ enhancement variability that may not be attributed to surface fluxes, which collapses the data pairs

more along the 1:1 line (compare Figs. 5b and 5c).

Therefore, our tests with CarbonTracker model reanalysis reveal that $XCO_2$ can indeed be used to viably estimate monthly surface $CO_2$ fluxes with simple mass balance approaches, especially over spatial extents of natural ecosystems. However, the method requires favorable conditions mainly related to advection. Namely, the region size must be large enough to account

for atmospheric mixing that could dominate transport in smaller domains over monthly timescales. Additionally, based on Figures 4 and 5 and assumptions of the mass balance model, winds must flow consistently through the region with a similar direction. Given the need for $XCO_2$ enhancements, the transport should originate from the same background source region within a given month rather than from multiple background regions. We speculate that the method may additionally work well in the Western US given the upwind Pacific Ocean region tends to have low anthropogenic sources and relatively lower $CO_2$

surface emissions and anomalies altogether. Thus, the $XCO_2$ enhancement variability will likely not be dominated by the background region's $XCO_2$ variability.

While the simple mass balance approach appears suitable for use based on a model reanalysis framework, repeating the procedure with observations such as with OCO-2 presents additional challenges, such as with observation error and spatiotemporal coverage. As such, CarbonTracker performance here effectively serves as an upper bound on predicting $XCO_2$'s ability to be coupled to surface $CO_2$ fluxes, acknowledging modeling sources of error. We address these issues in the following section.

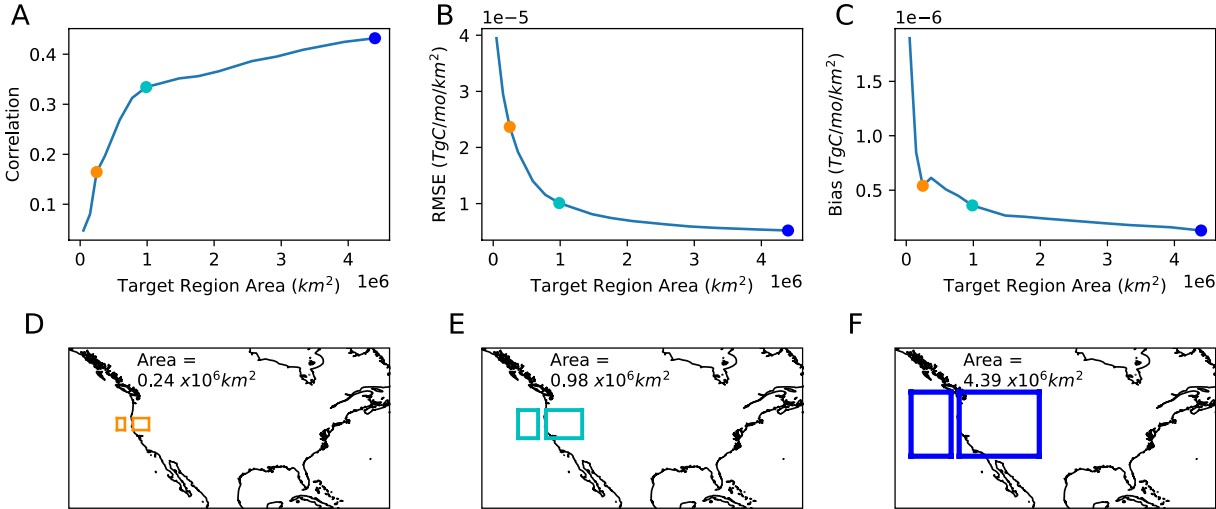

**Figure 3.** Performance of the $XCO_2$-based $CO_2$ flux estimation varying the target domain area. **(a)** Correlation, **(b)** root mean square error, and **(c)** bias between the CarbonTracker monthly surface flux anomaly outputs (considered here as the reference) and the $CO_2$ flux anomaly estimation with Eq. 1 based on $XCO_2$ from CarbonTracker. **(d, e, f)** Target domain sizes and locations where their domain border colors match the dot symbol colors in panels A-C.





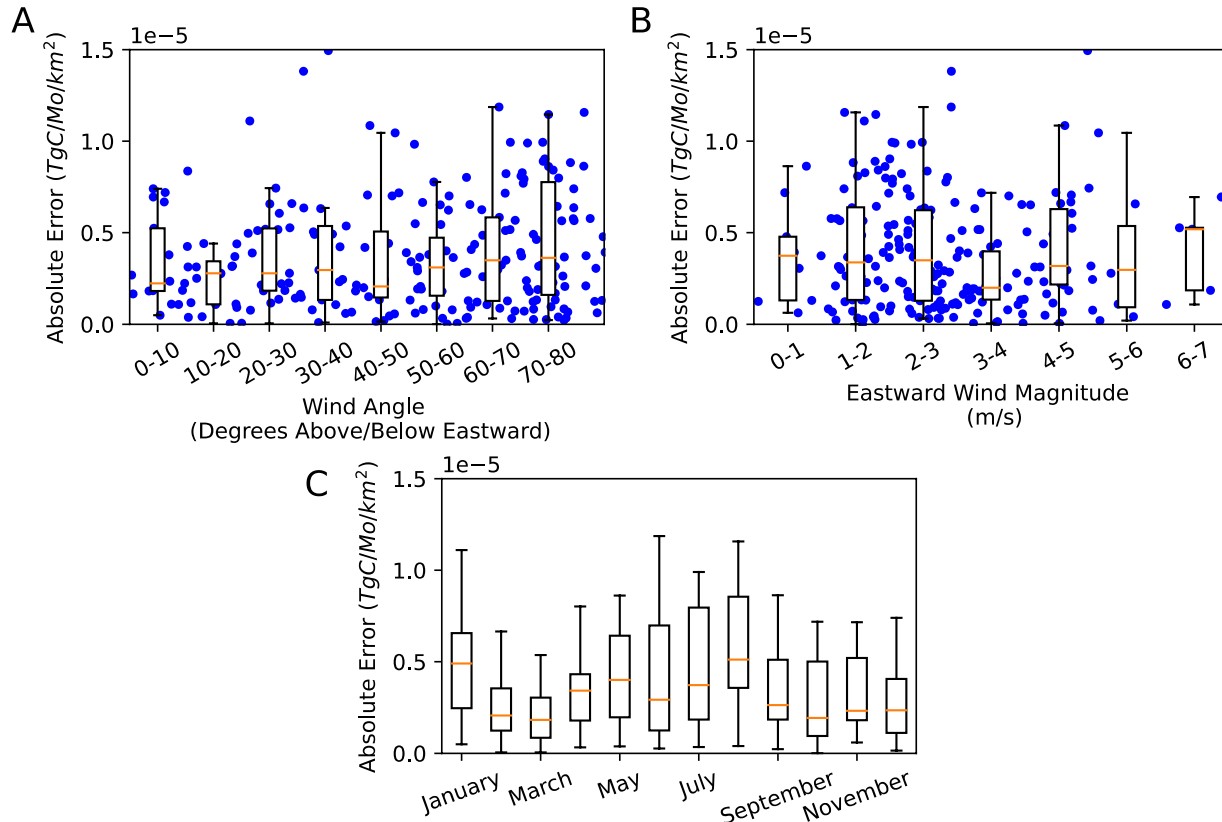


**Figure 4.** Effect of monthly averaged horizontal ventilation wind conditions on carbon flux anomaly estimation using CarbonTracker outputs. Carbon flux anomaly estimation error with respect to boundary layer **(a)** wind angle and **(b)** wind speed. **(c)** Carbon flux anomaly estimation error averaged over each month of year. Absolute error is the absolute value of the difference between each pair of CarbonTracker $XCO_2$ flux estimates using Eq. 1 and CarbonTracker surface $CO_2$ flux outputs.

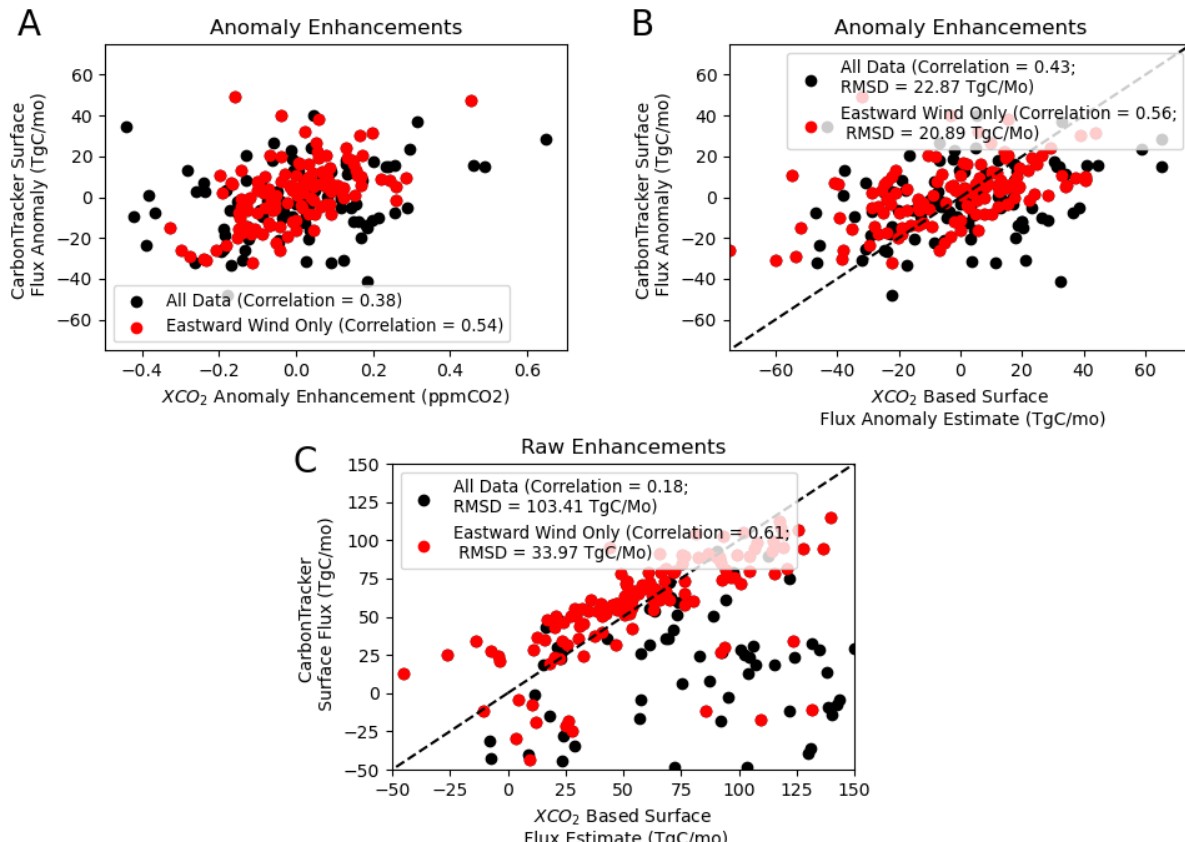


**Figure 5.** CarbonTracker XCO₂ flux estimation overall performance in the Western US considering a spatially expansive target domain (latitude = 33° N - 49° N, longitude = 124° W - 104° W as shown in Fig. 3f). Only CarbonTracker data was used here where its XCO₂, wind velocity, and pressure outputs were used to estimate surface CO₂ fluxes with Eq. 1, which are compared to CarbonTracker total surface CO₂ flux outputs. **(a)** Relationship between CarbonTracker-output surface CO₂ flux anomalies and CarbonTracker XCO₂ anomaly enhancements.

**(b)** Relationship between CarbonTracker surface CO₂ flux anomaly outputs and mass balance-based surface CO₂ flux estimates based on CarbonTracker XCO₂ anomaly enhancements. **(c)** Same as **(b)** but estimating raw total surface fluxes with total XCO₂ enhancements instead of anomalies. Legends show correlations and root mean square differences between the CarbonTracker XCO₂-based flux estimates (Eq. 1) and CarbonTracker surface CO₂ flux outputs. "Eastward Wind Only" includes only data pairs when the incoming wind direction from the Pacific Ocean is between -60º and 60º angles from eastward reference direction.

## 415 3.3 Observations Evaluation

### 3.3.1 OCO-2 XCO₂ Anomaly Coupling to Surface CO₂ Fluxes

OCO-2 XCO₂ anomalies are coupled to biospheric flux anomalies in the Western US region as expected from the CarbonTracker analysis in Sect. 3.2 (Fig. 6). XCO₂ anomalies negatively correlate with land surface model simulated net biome production and satellite-derived gross primary production anomalies, especially in the Western US domain (Fig. 6). For





example, monthly carbon uptake from increases in NBP and GPP are coincident with decreases in $XCO_2$ and vice versa. Their coupling is less consistent and reduced in the East US, presumably due to less ideal advection conditions where transport anomalies confound direct surface-atmosphere $CO_2$ coupling. Nevertheless, the Western US surface-atmosphere $CO_2$ coupling appears weaker than that shown in Fig. 5, which may be due to several additional sources of error that we investigate in Sect. 3.3.2.


The fact that spatially averaged $XCO_2$ anomalies are coupled to surface fluxes (Fig. 7) suggests that observed $XCO_2$ shows promise for directly detecting and estimating large-scale biospheric surface fluxes without the use of land surface and atmospheric transport assimilation models. The $XCO_2$ coupling to land surface carbon fluxes tends to increase when Pacific Ocean background $XCO_2$ are subtracted from Western US $XCO_2$ (Fig. 7a) (i.e., when $XCO_2$ anomaly enhancements are used).

The coupling further increases when only months with eastward flowing winds into the region are considered, at least for the total flux estimates (Fig. 7a). Such increased coupling accounting for advection in this way is expected from the CarbonTracker model reanalysis tests (Fig. 5a). This is because $XCO_2$ anomalies coupling with surface $CO_2$ fluxes are confounded by $XCO_2$ change due to transport conditions. For example, many Western US $XCO_2$ anomalies appear correlated to Pacific Ocean background $XCO_2$ anomalies in 2015 to 2016 suggesting that Western US $XCO_2$ variations were dominated by atmospheric

transport rather than surface fluxes in this time period (Fig. 7c). Therefore, $XCO_2$ anomaly enhancements can increase this coupling to the surface $CO_2$ fluxes by removing the confounding variations of background Pacific Ocean $CO_2$ concentrations transported into the region (Fig. 7d). However, some large anomaly enhancements may occur in months that advection was not consistently flowing through the region, thus requiring conditioning on wind angles. This removes cases of large $XCO_2$ differences between the two regions that may not necessarily be enhancements because atmospheric transport between these

two regions was reduced in these months (examples can be seen in late 2017 in Figs. 7c and 7d).

Even after isolating the effects of background Pacific Ocean $XCO_2$ and abnormal advection conditions, the magnitude of these observation-based correlations of 0.32 (Fig. 7a) are lower than that of CarbonTracker reanalysis tests at a correlation of 0.54 (Fig. 5a). Indeed, the total fluxes are estimated by LPJ NBP and QFED burning biomass which include model estimations and

assumptions with their own sets of errors. However, the surface-atmosphere carbon coupling is similar considering only photosynthesis fluxes from independently estimated GPP (Fig. 7a), which suggests a large role of the biosphere on the carbon fluxes and that LPJ model error may not be the main contribution to the correlation reduction. We ultimately expect that a main source of reduction in coupling originates from OCO-2 measurement and retrieval error.





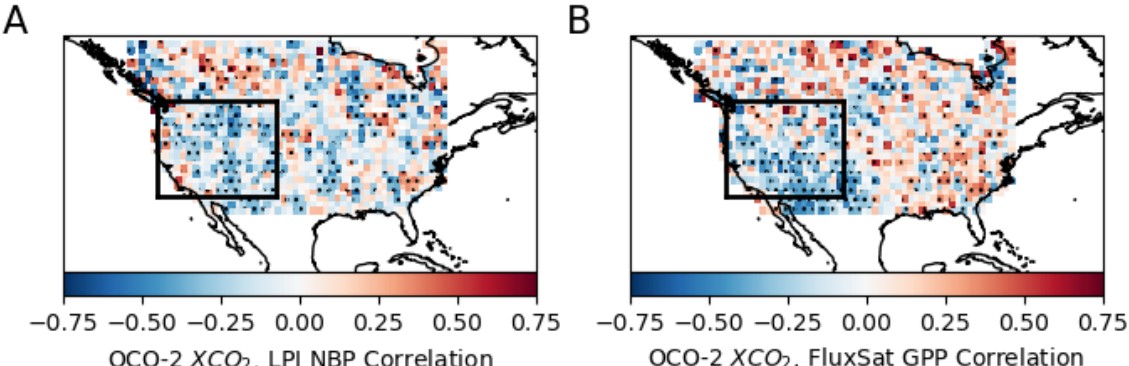

**Figure 6.** Observed OCO-2 $XCO_2$ anomalies are correlated with independent biospheric surface fluxes in Western US (with no time lag). **(a)** Correlation between OCO-2 $XCO_2$ anomalies and LPJ modeled net biome production anomalies. **(b)** Correlation between OCO-2 $XCO_2$ anomalies and observation-based FluxSat gross primary production anomalies. Negative correlations indicate that $XCO_2$ anomalies are coupled with the biospheric surface anomalies (i.e., increases in biosphere carbon uptake result in decreases in $XCO_2$). Pixels with stippling indicate p values less than 0.1.





**Figure 7. (a)** Pearson correlation coefficients between the $XCO_2$ anomalies and total surface $CO_2$ flux estimates (LPJ model and QFED biomass burning) as well as with observation-based FluxSat (** p-value<0.05; * p-value<0.1). **(b)** Map of Western US and background Pacific Ocean background domain definitions. **(c)** Spatially averaged OCO-2 $XCO_2$ anomalies in the Western US and background Pacific Ocean. **(d)** Western US $XCO_2$ anomaly enhancements from the background Pacific Ocean OCO-2 $XCO_2$ anomalies. Red symbols are months when the incoming wind direction from the Pacific Ocean was between -60° and 60° angles from eastward reference direction.

### 3.3.2 Error Estimation of $XCO_2$ Anomaly Enhancements

While subtracting a pair of $XCO_2$ anomalies to obtain anomaly enhancements increases errors above that of individual OCO-2 $XCO_2$ retrievals, spatial averaging of $XCO_2$ across a large domain reduces these errors (Fig. 8). OCO-2 $XCO_2$ error standard deviations for individual sounding retrievals are estimated to be 0.6 ppm in the Western US and 0.55 ppm in the Pacific Ocean. This results in an error standard deviation of approximately 0.82 ppm for $XCO_2$ anomaly enhancements (or difference between a pair of $XCO_2$ retrievals between the Western US and Pacific Ocean) (Fig. 8a). However, assuming normally distributed and independent errors, spatial averaging of approximately 20 observations within the Western US target region and 10




observations in the Pacific Ocean background region (typical OCO-2 observation counts in the study region in Fig. 7B) reduces this error standard deviation of Western US $XCO_2$ anomaly enhancements to around 0.2 ppm (Fig. 8b). This error magnitude is similar to that found in Chatterjee et al. (2017) when averaging over monthly timescales over a region. We find that spatial autocorrelation of errors (relaxing assumption of independent errors) may not greatly change this error standard deviation due to competing effects (Fig. 8c). Specifically, spatial autocorrelation of errors removes some noise reduction benefits when

averaging within a region due to a spatial relationship between the errors, rather than random errors. However, this spatial relationship in errors also results in a partial canceling of errors when subtracting the spatially averaged Western US $XCO_2$ anomaly error from the spatially averaged Pacific Ocean $XCO_2$ anomaly error to obtain $XCO_2$ anomaly enhancements. Ultimately, a given month's $XCO_2$ anomaly enhancement in our Western US target region is about 0.2 ppm, or a third of that of a single $XCO_2$ retrieval's error. Therefore, aggregating $XCO_2$ anomalies and enhancements spatially and temporally may

allow detection of smaller $XCO_2$ anomalies given that the emission source anomalies themselves are on the order of the spatial and temporal aggregation.

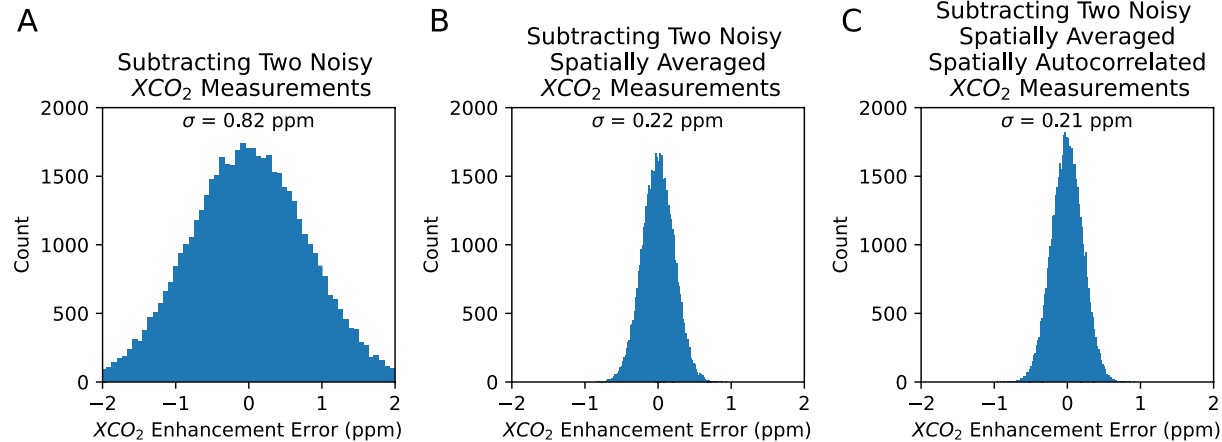

**Figure 8.** $XCO_2$ enhancement error based on subtracting two noisy, spatially averaged $XCO_2$ retrievals using simulated additions of random $XCO_2$ error. For a given sounding, OCO-2 $XCO_2$ retrieval error is on average 0.6 ppm in the West U.S. and 0.55 ppm in the Pacific Ocean.

**(a)** $XCO_2$ enhancement error considering subtracting two independent, noisy measurements. **(b)** $XCO_2$ enhancement error considering the spatial averaging of $XCO_2$ retrievals in the target and background regions, with all errors assumed to be independent. **(c)** $XCO_2$ enhancement error considering the spatial averaging of $XCO_2$ retrievals in the target and background regions, but assuming errors are spatially autocorrelated within each region and spatially autocorrelated between the background and target region.

### 3.3.3 OCO-2 $XCO_2$ Estimation of Monthly Surface $CO_2$ Fluxes

Carbon flux anomaly estimates from OCO-2 $XCO_2$ using Eq. 1 weakly co-vary with modeled and observation-based surface $CO_2$ flux anomalies (Fig. 9). In general, the simple mass balance method increases its ability to estimate surface $CO_2$ fluxes when conditioning on the "best" atmospheric transport conditions as shown across correlation, mean bias, and RMSD statistics





(Fig. 9). However, the performance of the flux estimation method is reduced overall when using OCO-2 observations compared to CarbonTracker model reanalysis tests (shown for comparison in Fig. 9).


Even though spatial averaging reduces observation error of XCO$_2$ anomaly enhancements, we show that reduced simple mass balance flux estimation performance can largely be attributed to OCO-2 XCO$_2$ retrieval errors (from OCO-2 instrument measurement error and algorithmic retrieval error). Specifically, adding this approximate 0.2 ppm error randomly to CarbonTracker XCO$_2$ outputs results in comparison statistics that approach that based on observed OCO-2 (see Figs. 9b to 9d

especially for correlation and RMSD). Other error sources likely also explain the reduced comparison between OCO-2-based estimates and surface modeled estimates including MERRA2 wind vector error, reference surface flux error (from LPJ biosphere model and QFED fire estimate error), inconsistent XCO$_2$ spatiotemporal coverage within each region, and Eq. 1 mass balance model errors. However, our test reveals that greenhouse gas satellite retrieval error is a dominant component of the overall error in estimating surface fluxes. Ultimately, the retrieval error in OCO-2 XCO$_2$ hinders reliable estimation of

nominal monthly surface flux anomalies using rapid mass balance approaches, as expected from simulations (Chevallier et al., 2007). Indeed, more accurate greenhouse-gas satellite missions would be needed to approach the potential surface flux estimation performance with mass balance models as suggested by CarbonTracker reanalysis.

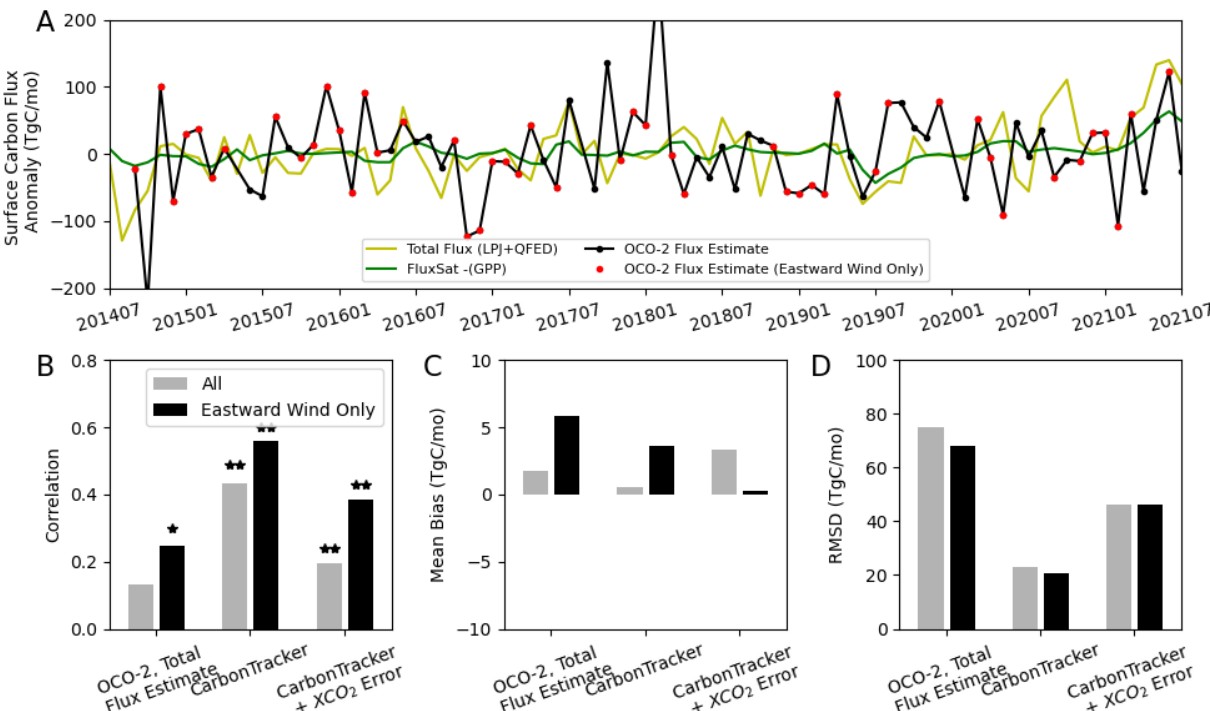

**Figure 9. (a)** Spatially averaged OCO-2 XCO$_2$ flux anomaly estimates compared to total flux estimate anomalies (LPJ model and QFED

biomass burning) and FluxSat gross primary production anomalies. Positive anomalies are fluxes away from the surface. Comparison





statistics between OCO-2 flux anomaly estimates and LPJ NBP anomalies with **(b)** correlation (** p-value<0.05; * p-value<0.1), **(c)** mean bias, and **(d)** root mean square difference (RMSD). CarbonTracker comparisons are shown for reference repeated from Fig. 5. CarbonTracker+$XCO_2$ Error includes simulated error added to CarbonTracker modeled $XCO_2$ on the order of that shown in Fig. 8. The statistics are computed for all data pairs as well as only those considering months when the incoming wind direction from the Pacific Ocean
was between -60º and 60º angles from eastward reference direction.

### 3.3.4 OCO-2 $XCO_2$ Detection and Estimation of Extreme Surface $CO_2$ Fluxes

Although OCO-2 measurement noise limits estimation of smaller monthly surface flux anomalies using $XCO_2$, OCO-2 $XCO_2$ retrievals show promise in directly detecting and estimating the largest surface $CO_2$ flux anomalies. Despite OCO-2 noise levels (of 0.2 ppm to 0.6 ppm depending on averaging of individual soundings), large $XCO_2$ anomalies above the noise are
likely indicative of a large surface flux anomaly in the Western US. This is expected from previous studies that $XCO_2$ 0.5 ppm changes or greater may be detecting a physically-driven atmospheric carbon concentration anomaly (Chatterjee et al., 2017).

To highlight the ability of $XCO_2$ anomaly enhancements to directly detect extreme carbon cycle events in the Western US, we evaluate the $XCO_2$ detection rate (Eq. 2) of surface $CO_2$ anomalies across a range of nominal to extreme conditions (Fig. 10).
Figs. 10a to 10C show that when $XCO_2$ anomaly enhancements are large (>90th percentile), they are frequently detecting above average surface $CO_2$ flux anomalies for both observation and model assessments. As expected from Figs. 8 and 9, the detection rates are higher in the CarbonTracker reanalysis testbed across all conditions than for observations, likely due to OCO-2 $XCO_2$ retrieval error considerations. Additionally, as expected from positive correlations between surface $CO_2$ flux anomalies and $XCO_2$ anomaly enhancements, larger $XCO_2$ anomaly enhancements are better able to detect surface $CO_2$ flux anomalies than
smaller $XCO_2$ anomaly enhancements. Figs. 10d to 10f specifically show the degree to which an $XCO_2$ anomaly enhancement can detect a given magnitude of surface $CO_2$ flux anomalies greater than by chance. CarbonTracker $XCO_2$ anomaly enhancements can detect surface carbon flux anomalies of at least the same percentile well above that by chance in nearly all cases (Fig. 10f). However, OCO-2 $XCO_2$ anomaly enhancements can only detect surface $CO_2$ flux anomalies when $XCO_2$ anomaly enhancements are the largest (>90th percentile), demonstrating how OCO-2 retrieval error largely removes the surface
flux information content of a smaller magnitude $XCO_2$ anomaly (Figs. 10d and 10e). Nevertheless, the largest OCO-2 $XCO_2$ anomalies enhancements within the Western US are frequently associated with a surface $CO_2$ flux anomaly.

In the context of terrestrial biosphere extremes (i.e., droughts and heatwaves), we additionally evaluate whether extreme surface $CO_2$ efflux anomalies create a positive $XCO_2$ anomaly (Fig. 11). Using OCO-2 $XCO_2$ anomalies alone (without
enhancements), the $XCO_2$ detection rate of the largest biospheric surface efflux anomalies (>95th percentile) exceeds 50% (Fig. 11). Thus, OCO-2 will detect the surface $CO_2$ flux signal as a positive $XCO_2$ anomaly under extreme biosphere conditions, beyond only by chance. OCO-2 $XCO_2$ anomalies detect 60% of the largest total surface efflux anomalies in the Western US study domain (Fig. 11a). With reduced satellite instrument noise, the detection could increase to 80%, a detection rate potential



estimated from CarbonTracker. When considering photosynthesis anomalies available globally with the MODIS-based
FluxSAT GPP dataset, XCO$_2$ positive anomalies have detected between 60% and 100% of the most extreme negative
photosynthesis anomalies across several regions of the world (Fig. 11). These other regions, including the Sahel, Central
Europe, and West Australia, were selected given that they have had extreme biospheric anomalies over the past two decades
(Fig. 11). In assessing only the XCO$_2$ anomalies, we are not considering transport conditions, where regions like West Australia
have transport conditions less conducive for surface flux estimation with Eq. 1 than the Western US. Nevertheless, when a
climatic event is ongoing and model outputs of surface carbon fluxes are not yet available, OCO-2 XCO$_2$ anomalies can be
rapidly consulted. If a large XCO$_2$ anomalies is detected, it can be used as the motivation to initiate a more detailed
investigation and/or monitoring campaign of the climatic event.

We also show that the simple mass balance method (Eq. 1) approximately estimates these extreme fluxes that it detects in the
Western US (Fig. 12). The 2021 fluxes in March and June were part of an extreme Western US drought and heatwave event
(Philip et al., 2021; Williams et al., 2022). The LPJ model and QFED wildfire estimates indicated that these total efflux
anomalies increased to a peak in Spring 2021 (Fig. 9a). In June, the OCO-2-based flux estimate finds a 122 TgC/mo anomaly,
while the independent total flux estimate from LPJ and QFED is 140 TgC/mo (Fig. 12). Therefore, the simple mass balance
method provides a viable method to rapidly estimate the extreme fluxes from a satellite observation source compared to more
complex modeling and reanalysis. In the months when XCO$_2$-based flux anomalies did not compare with that estimated in
2020, the total flux estimates (from LPJ and QFED) potentially were positively biased when FluxSat GPP did not indicate
large biosphere flux anomalies (Fig. 9a). Therefore, the extreme flux estimates from other sources may have had model related
errors that resulted in the reduced comparison.

While such a simple mass balance approach does not supplant a rigorous flux assimilation, it serves as a rapid estimation
approach that can be used within one to two months latency. This is significant given the typical one-month latency of
greenhouse gas satellites like OCO-2, while total surface flux estimates that require biosphere model ensemble
implementations are often multi-month or multi-year efforts. As such, greenhouse gas satellites can be consulted for rapid
monitoring and attribution to determine whether an ongoing extreme climatic anomaly (i.e, the Western US 2020-2021
drought) is creating substantial carbon cycle anomalies.



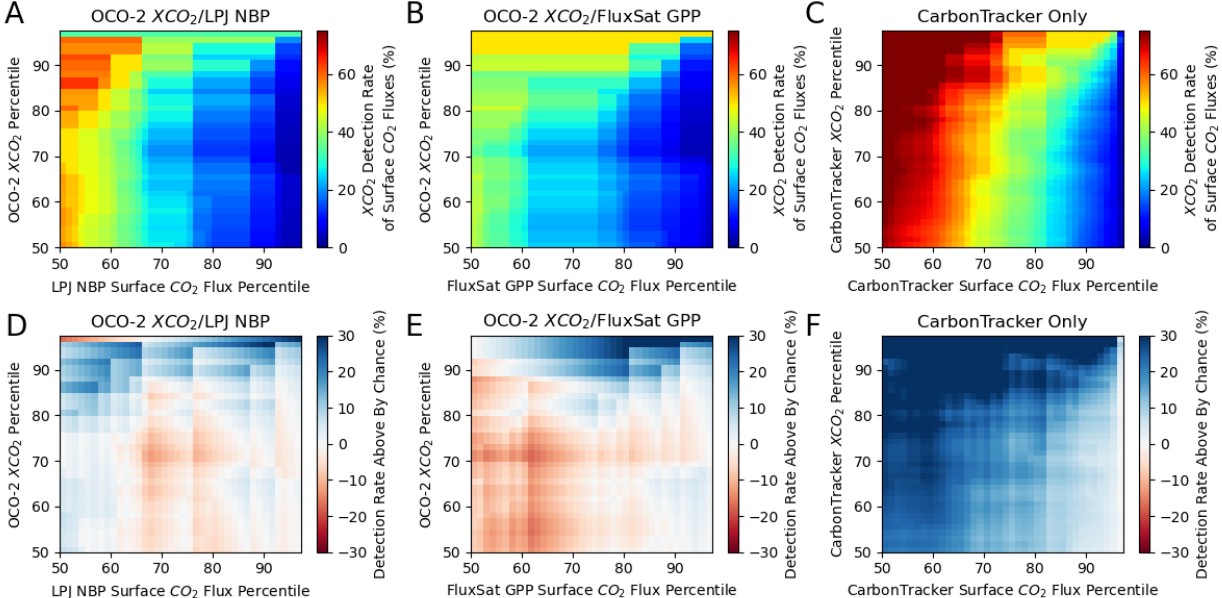

**Figure 10.** OCO-2-retrieved $XCO_2$ anomaly enhancements that are more extreme (>90th percentile) can detect surface $CO_2$ anomalies. By contrast, less extreme retrieved $XCO_2$ anomaly enhancements (50th-80th percentile) have little ability to detect surface $CO_2$ anomalies. **(a, b, c)** Western US observed $XCO_2$ anomaly enhancements detection rate of surface $CO_2$ fluxes for **(a)** OCO-2 $XCO_2$ detection of LPJ NBP surface fluxes and **(b)** OCO-2 $XCO_2$ detection of FluxSat GPP surface fluxes. This is compared to detection rates from the reanalysis testbed in the absence of satellite retrieval error for **(c)** CarbonTracker $XCO_2$ detection of CarbonTracker $CO_2$ surface fluxes. Each detection rate value is estimated by binning all $XCO_2$ anomaly enhancements above the given percentile (y-axis) and determining the number of coincident monthly surface $CO_2$ flux anomalies that are above the given $CO_2$ flux percentile (x-axis). Detection rates are computed based on Eq. 2. **(d, e, f)** Same as **(a, b, c)** but subtracting the rate of detection by chance. Values that are positive (blue) indicate that $XCO_2$ anomaly enhancements are better able to detect surface fluxes than by chance.





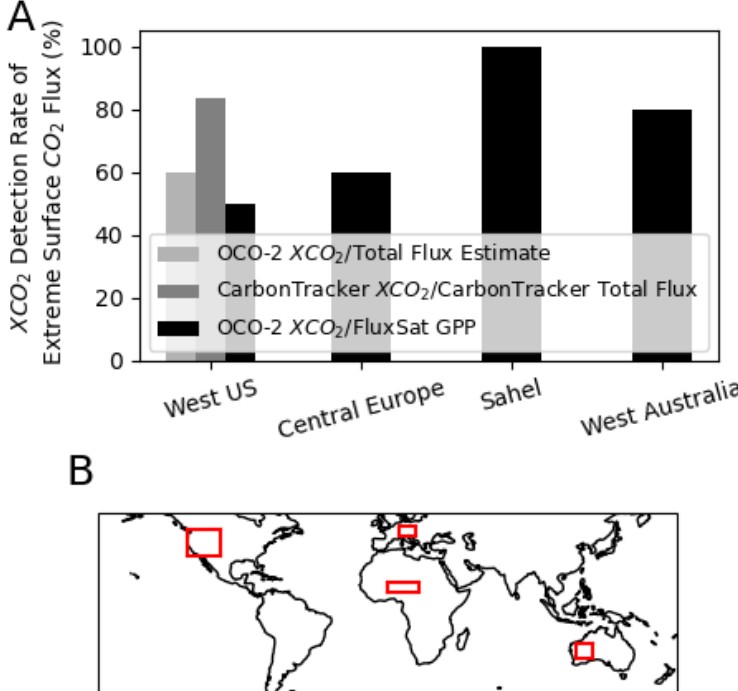

**Figure 11.** OCO-2 XCO2 anomalies alone can detect extreme CO2 fluxes. **(a)** OCO-2 detection rate of extreme surface fluxes (or positive
XCO2 anomalies when surface flux anomalies are the largest). The legend shows the datasets used to estimate the detection rate including
pairs of OCO-2 XCO2 anomalies and total flux estimates from LPJ and QFED, pairs of CarbonTracker XCO2 anomalies and CarbonTracker
total flux estimates, and pairs of OCO-2 XCO2 anomalies and photosynthesis flux estimates from FluxSat GPP. **(b)** Reference map of regions
shown in **(a)**.

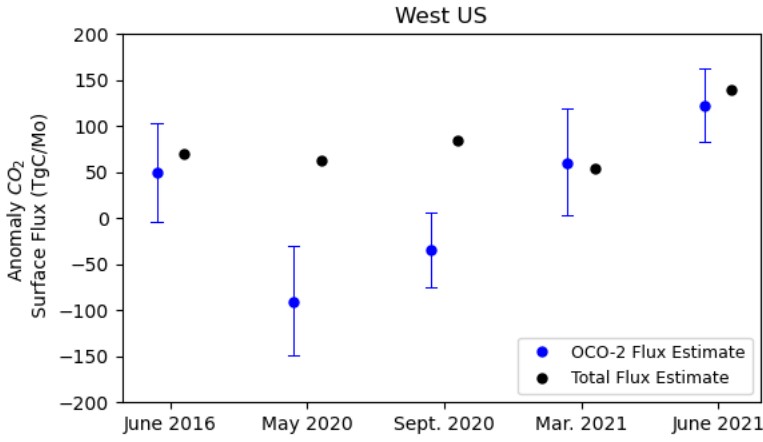







**Figure 12**. OCO-2 can roughly estimate extreme surface $CO_2$ fluxes. OCO-2 estimation of extreme surface fluxes in the Western US target domain. Total flux estimates are surface fluxes estimated from a combination of a dynamic global vegetation model (LPJ) and wildfire model reanalysis estimates. Error bars are determined from bootstrapping error estimates determined in Fig. 8c. Fossil fuel anomalies are negligible in magnitude compared to the biosphere and fire sources (see Fig. S1).

**4 Conclusions**

We demonstrate that OCO-2 satellite retrieved $XCO_2$ can be used with mass balance frameworks to detect and estimate biospheric $CO_2$ flux anomalies at monthly timescales, which exceeds expectations of such greenhouse gas satellites. The application tested here ultimately requires aggregating $XCO_2$ over regional spatial domains with careful consideration of transport conditions. Namely, the surface flux estimation mass balance method using $XCO_2$ improves when using larger spatial domains and when wind conditions are on average from the same background location in a given month and flow consistently through the target domain. The larger spatial domain reduces errors due to turbulent atmospheric mixing of surface $CO_2$ sources that would hinder use of a source pixel mass balance method. Additionally, use of the larger area inherently requires aggregation of several $XCO_2$ soundings which reduces the magnitude of $XCO_2$ errors.

Satellite $XCO_2$ anomalies from OCO-2 are particularly useful for evaluating more extreme biosphere fluxes. We show here that the timing and magnitudes of extreme $CO_2$ fluxes can be monitored where OCO-2 $XCO_2$ instrument and retrieval error hinder evaluating smaller flux anomalies. In the absence of this error, the performance of these methods greatly improve as suggested by CarbonTracker reanalysis. Therefore, any reduction in $XCO_2$ measurement and retrieval error in upcoming greenhouse gas missions (i.e., GeoCarb) may extend the ability to globally monitor the timing and magnitude of biosphere anomalies at shorter timescales (beyond that of their design specifications to evaluate aggregated carbon cycle responses at longer than seasonal timescales). Furthermore, even if advection conditions prevent use of the simple pixel source mass balance method, extreme fluxes at least can be detected using only the observed monthly $XCO_2$ anomaly within the target domain. In addition to the Western US study domain here, this anomaly-only approach is demonstrated in other domains like the Sahel, Europe, and Western Australia that may have more complex advection conditions.

The value of such a means to monitor and estimate total surface carbon fluxes is manifold: it is simple in not requiring many assumptions and ancillary datasets; it is rapid and therefore can be used as a first estimate in monitoring extreme events; it uses $XCO_2$ which integrates all surface $CO_2$ flux sources, the components of which otherwise need to be estimated separately in bottom-up approaches; since it is based mainly on observations independent of land surface models, it can be used as independent estimate to evaluate global model surface carbon flux outputs.



We recommend that future work determine all global terrestrial ecosystems with favorable wind conditions as defined in the study where the surface carbon flux estimation method can be applied. A particular focus should be placed on tropical locations where measurement networks are sparse with consequently uncertain model outputs as well as where the biosphere takes up 625 the largest proportion of anthropogenic emissions. These methods should also be developed for near real time monitoring in known climate change hotspot regions, as is done here for the Western US hotspot, where more frequent and intense climate anomalies are expected in the future.

## 5 Code/Data Availability

All datasets used here are freely availability. CarbonTracker reanalysis data (CT2019B) are available at 630 https://gml.noaa.gov/ccgg/carbontracker/. MERRA2 reanalysis wind and pressure fields are available at https://disc.gsfc.nasa.gov/datasets/M2T3NVASM_5.12.4/summary. OCO-2 $XCO_2$ retrievals and QFED outputs are available at https://disc.gsfc.nasa.gov. The LPJ model code is publicly available at https://github.com/benpoulter/LPJ-wsl_v2.0.

## 6 Author Contributions

B.P. conceived the study. A.F.F. conducted the analysis and wrote the manuscript. B.P. and A.C. led the study. Z.Z. ran the 635 LPJ model. Y.S. provided GPP remote sensing retrievals and offered guidance on their interpretation. All authors contributed interpretations of figures and textual edits.

## 7 Acknowledgements

Andrew F. Feldman's research was supported by an appointment to the NASA Postdoctoral Program at the NASA Goddard Space Flight Center, administered by Oak Ridge Associated Universities under contract with NASA.

## 640 8 Competing Interests

The authors declare that they have no conflict of interest.

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
