# Peer review of "Using OCO-2 column CO2 retrievals to rapidly detect and estimate biospheric surface carbon flux anomalies"

_Atmospheric Chemistry and Physics, 2022_

## Author Response (AR1)

Dear Dr. Christoph Gerbig,

My co-authors and I are excited to submit revisions to our manuscript, "Using OCO-2 column $CO_2$ retrievals to rapidly detect and estimate biospheric surface carbon flux anomalies," to *ACP* for review. Here, we show that OCO-2 $XCO_2$ retrievals have value for rapidly evaluating ongoing extreme surface flux conditions, especially in the terrestrial biosphere.

We appreciate the constructive comments from both reviewers. We have responded to their comments in agreement, and we feel that their comments have greatly improved the manuscript. Ultimately, our main conclusions still stand. We summarize our main edits here.

In response to Reviewer 1, our main edits include a more comprehensive region selection analysis that assists in determining where conditions are met for atmospheric $CO_2$ concentration coupling to surface $CO_2$ flux anomalies. We also have also revised our analysis of $XCO_2$ anomaly error estimation and evaluated effects of vertical motion on our results.

In response to Reviewer 2, Dr. Prabir Patra, we have increased readability by reducing descriptive text throughout the manuscript. Notably, there was a reduction in word count in many sections despite new analysis/sections being added. Furthermore, we have reduced the number of figures from 12 to 9, with nearly a 50% reduction in the total number of panels.

Ultimately, these findings are of interest to both atmospheric remote sensing and terrestrial carbon cycle science communities in establishing conditions under which satellite-based XCO2 anomalies can be linked to regional surface carbon dioxide fluxes. As such, we believe our manuscript will be of interest to *ACP's* audience. We look forward to your response.

Sincerely,

Andrew Feldman
NASA Postdoctoral Program Fellow
Biospheric Sciences Laboratory, Goddard Space Flight Center
andrew.feldman@nasa.gov

**Important:** Please note that line and figure numbers in reviewer comments in the left column are in reference to the **initial submission**. The line and figure numbers in author response in the right column are in reference to the revised manuscript submission.

| Reviewer Comments | Response |
|---|---|
| **Comment 1:**
Feldman et al. present an analysis of the ability to use OCO-2 XCO2 observations to detect and estimate biospheric surface CO2 flux anomalies over the Western US using a simple mass balance approach. They find that in a synthetic testbed scenario using CarbonTracker estimates and a large enough domain to reduce the inflow of background CO2 concentrations the simple mass balance approach is capable of detecting monthly surface CO2 flux anomalies. However, in a real world scenario with OCO-2 XCO2 observations this method is only capable of detecting large surface anomaly enhancements and only when the OCO-2 XCO2 anomaly enhancements are above the 90th percentile.

This is a well written and structured manuscript exploring an interesting and alternative (to atmospheric transport inversions) application of the OCO-2 XCO2 observation. The readability and scientific credibility of the manuscript will benefit from a few clarifications by the authors. | We appreciate the constructive and thought-provoking comments. See our responses below. |
| **Comment 2:**
1. What constitutes the XCO2 retrieval noise level (mentioned in line 74), does that also include both systematic and random errors? Later on (lines 472ff), the authors argue that spatial autocorrelation of errors does not change their derived error standard deviation when relaxing the assumption of independent errors. This is not clear to me; there should be a difference of 1/sqrt(n), with n being the number of averaged observations, between assuming fully correlated errors and independent errors. Further, the authors mention compensating effects due to subtracting two anomaly error estimates (Western US XCO2 anomaly error minus Pacific Ocean XCO2 error anomaly), but this should rather increase the error of the difference. | Note that we have reduced mention of the error estimation because it is a subtle point of the study that the results do not greatly rely upon. We have only mentioned the error estimation briefly in the methods and moved the points in the introduction to the discussion section. Namely, our results stand whether the XCO2 enhancement error is estimated to be for example as low as 0.2 ppm or 0.6 ppm. See our new paragraph in lines 481-491.

$XCO_2$ retrieval error is estimated by the OCO-2 science team to be due to those related to measurement error and additional errors from algorithmic retrieval uncertainty (numerical estimation and algorithmic/radiative transfer assumptions) as confirmed by their ATBD document. See line 484. |

| | Our mention of spatial autocorrelation is mainly a caveat, but does not influence our results. To clarify our point, if there is spatial autocorrelation across a broader region, indeed averaging within the region will produce much smaller error reductions than if the errors are independent. However, the errors are also spatially positively autocorrelated between the background and target region as well. Therefore, when subtracting two averaged $XCO_2$ values to obtain the enhancement, there is a partial canceling of this error. The degree of spatial autocorrelation is not well known. We have rerun our tests and found cases where spatial autocorrelation of error increases $XCO_2$ enhancement error (from 0.2 ppm to 0.3 ppm).

Ultimately, we have de-emphasized these points because they do not change our conclusions about the ability for OCO-2 to detect large surface emissions. They are mainly used to make qualitative points about attributing $XCO_2$ flux estimations to $XCO_2$ retrieval error. Our new paragraph in lines 481-491 clarifies these points. |
|---|---|
| **Comment 3:**
2. The authors do consider the effect of advection of CO2 from background regions perturbing the signal in the XCO2 observations but they neglect the impact of inflow of CO2 to a total column estimate from atmospheric layers above the boundary layer. The study would be strengthened if the authors could show that this is negligible. | $XCO_2$ used in this study (and required by the model) does in fact include the full column - both $XCO_2$ data from the CarbonTracker reanalysis tests and the OCO-2 retrievals are full column integrations. See lines 216-221 that were revised to make this point clearer.

However, we have not discussed issues related to the effect of vertical wind velocity on the flux estimation in Eq. 1. Therefore, we have added an experiment that also assesses the vertical wind velocity on CO2 flux estimation error. See line 245. We found that vertical wind velocity does not have an apparent impact on our results here. See our added discussion in lines 382-389. |
| **Comment 4:**
3. How is the analysis impacted by the choice of region, especially since there has been an 'ongoing decadal-scale megadrought' and the XCO2 climatology only consists of less than a decade? | For our broader response to choice of the region, please see our response to comment 5 below.

We do not anticipate that the mean climatology being based on decadal drought conditions confounds the study. Our detection and estimation are based on the anomalies of $XCO_2$ (see lines 230-234). Therefore, having $XCO_2$ mean climatology being potentially higher than average for the decade (because of lower carbon uptake) results in positive anomalies being |

| | those on top of the background conditions. They are therefore of even more substantial magnitude than if the XCO$_2$ climatology were available for well before the early 2000s. We have added a sentence that reflects this in lines 314-315. |
|---|---|
| **Comment 5:**
4. What are the limiting factors for the selection of the domain? Or in other words which region characteristics influence the anomaly detection most: topography (and hence advection), heterogeneity in land cover, human footprint on the emissions in the domain, ….? | This is an excellent question. To address this question, we have made our region selection more rigorous by starting the results section with a new subsection section and figure (Figure 1). Namely, we use empirical metrics that quantify a link between the surface and atmospheric CO$_2$ and the quality of transport conditions that should integrate these physical considerations to motivate the region selection. See our new methods section in lines 167-195 and our new results section in lines 288-309. We have included two panels to Figure 1:

Panel A includes a monthly anomaly correlation between OCO-2 XCO2 and MODIS-based FluxSat GPP. It is an observation-based metric that shows the degree of direct coupling between biospheric surface fluxes and atmospheric carbon concentrations, and therefore indicates favorable transport conditions that don't confound this surface-atmosphere carbon link.

Panel B focuses on the MERRA2 wind direction variability and whether there are consistent wind conditions from favorable background sources. It does not directly say whether there is a link between biosphere fluxes and atmospheric carbon concentrations, but it does convey the potential for use of Eq. 1 mass balance approach.

These metrics allow an objective assessment of which regions are most conducive for using XCO$_2$ directly for obtaining information about surface CO$_2$ fluxes in the study. If both metrics are blue in Figure 1, it means advection conditions are tractable and biosphere fluxes over large areas are linked to the atmosphere, and therefore that a range of conditions are met given the physical constrains on our simple methods.

A more comprehensive analysis would be necessary to determine which of the factors that the reviewers mention (satellite instrument/algorithm versus land surface considerations) influence the estimation the most, which indeed is a very interesting question. For our purposes here, potentially all of these conditions |

| | need to be met (i.e., necessary conditions) to create a link between the surface and atmosphere to some degree (Fig. 1a). We note these points explicitly in lines 305-309. Nevertheless, the two metrics shown in Fig. 1 integrate these effects to determine where direct use of $XCO_2$ retrievals to understand surface $CO_2$ fluxes is possible. |
|---|---|
| **Comment 6:**
Some additional points:
L 90: Please add 'CO2' here: … can be used for surface CO2 fluxes - … | Note that this specific paragraph has been moved and this phrase removed in our revisions. However, we have added "CO2" to describe the fluxes throughout the manuscript (mainly replacing "carbon" which is ambiguous here). See line 68 for example. |
| **Comment 7:**
L 164/165: If the LPJ annual fire emissions and the annual sum of the QFED biomass burning emissions are not of the same size, this then effects the carbon closure in LPJ, i.e. the model would not be mass conserving anymore and LPJ would simulate more/less heterotrophic respiration in the following year (depending on the sign of the difference). I doubt that this effect changes the analysis in the manuscript but it is worth mentioning. | The concern is noted, which would indeed create a closure issue if done within the LPJ model run. However, this was done in postprocessing of the model outputs and did not influence the model simulation itself. Thus, there is no impact on the LPJ model run. See our revision in line 251-254. |
| **Comment 8:**
L184: Spatially averaged to which resolution? | We average all pixels within the target region between 33° N and 49° N latitude and 124° W and 104° W longitude. We have revised lines 207 to reflect this point. |
| **Comment 9:**
L 354/355: add 'be': … during the summer months may be the cause… | Thank you for catching this. This change has been made in line 374. |
| **Comment 10:**
L 635: Should it be 'Y.Y. provided GPP…'? | Yes, thanks for catching this error. See line 628. |
| **Comment 11:**
Fig 1: It would be nice to see each month individually and not seasonally averaged, at least as a supplemental figure. | We have added figure S2 which includes the monthly averaged wind quivers over each of the 12 months. Note that Figure 2A is now the averaged wind quiver plot in all years and seasons. |
| **Comment 12:**
Fig 4: How large are the errors in relative terms? | We have converted to a relative error by dividing the difference (between the CarbonTracker CO2 flux anomaly output and mass balance estimated CO2 flux anomaly) by the CarbonTracker CO2 flux output standard deviation. See updated Figure 4. Relative error values above 1 are therefore above one standard deviation. |
| **Comment 13:** | Please note that figure 6 has been removed. Figure 1A is now old Figure 6B, but is shown globally in |

| | |
|---|---|
| Fig 6: The stipples are not clearly visible, please revise such that it becomes clearer which gridcells show significant correlations. | our response to Comment 5 above. Therefore, we apply this comment to Fig. 1. Instead of stipples, we have chosen to make pixels transparent that aren't statistically significant. See Figure 1A. Stipples/hatching tended to crowd the image. |

**Responses to Reviewer #2**

**Important:** Please note that line and figure numbers in reviewer comments in the left column are in reference to the **initial submission**. The line and figure numbers in author response in the right column are in reference to the revised manuscript submission.

| Reviewer Comments | Response |
|---|---|
| **Comment 1:**
This paper tries to estimate CO2 fluxes based on a pure observational based system, using OCO-2 measurements of XCO2 and basic meteorological measurements from reanalysis. The CO2 flux calculation (divergence) method is similar to that has been applied to NO2 measurements from space commonly for estimation of NO2 emissions from hotspots. One major difference between CO2 and NO2 systems of flux derivation is the data density and the interference from land biosphere fluxes, with peculirities arising from the lifetimes of the two species of concern. | We appreciate the constructive comments. See our responses below. |
| **Comment 2:**
The manuscript is overly descriptive, and was very difficult to read for me. I have marked a few minor things on the PDF, but those I think not so important to discuss if the present form or anything close to this would be accepted for publication. | We have reduced and simplified the writing in many locations resulting in ~5% reduction in total word count even with new analysis and sections added:
1) Abstract: Simplified and reduced.
2) Introduction: Simplified and reduced. For example, the motivation was rewritten and reduced (i.e., lines 33-57). The detailed XCO$_2$ errors paragraph was moved to the discussion in lines 537-548.
3) Methods: Descriptive sections of the methods were reduced by removing material we deemed non-essential. This resulted in about 25% of the original text being removed.
4) Results: The text was greatly reduced where possible. For example, the original section on estimation of XCO2 enhancement error was removed and included as only a short paragraph in Section 3.3.2 in lines 481-491.

We have also reduced figures and panels from 12 to 9 total figures and 41 to 23 total panels:
1) Figures 1 and 2 (7 panels total) were consolidated into two panels and are new Figure 2. The remainder of panels were moved to the SI as Figs. S3 and S4.
2) Figure 3 number of panels was consolidated from 6 to 4.
3) Figure 5 was reduced to only 1 panel. The other 2 panels were moved to SI as Fig. S6. |

| | 4) Figure 6 was removed. New Figure 1 includes these contents. |
| | 5) Figure 8 was removed with the essential numbers stated in the text. |
| | 6) Figure 10 was moved to the SI. |

| **Comment 3:** | |
| As the authors have acknowledged it is very difficult to separate the influences of far and near fields on $CO_2$ flux estimation based on different area consideration in Fig. 3, application of divergence methods probably remained skeptical for $CO_2$ research given the data sensity and data quality of $CO_2$ (as mentioned earlier large difference in signal-to-noise ratios for $CO_2$ and $NO_2$ due to lifetimes), unless probably focussing at a hotspot. | We agree with this point. We have clarified our finding about $CO_2$ and its mixing and lifetime within a month in our region in lines 354-369. Ultimately, too small of a domain size with $CO_2$ over monthly timescales will result in greater errors. |

| **Comment 4:** | We may be misunderstanding this comment. LPJ outputs are used as a reference to compare with the observed OCO2 XCO2 retrieval anomalies. CarbonTracker is not compared to LPJ or observations here. Instead, CarbonTracker is used as a simulation space (though imperfect) to compare mass balance surface carbon flux anomaly estimates using CT's $XCO_2$ outputs to CarbonTracker's carbon flux outputs, which are considered the true flux values in this exercise. If the CarbonTracker modeling is perfect, then errors between the carbon flux outputs and those from the mass balance should be attributed to mass balance model assumptions. See lines 143-164 where these datasets are partitioned in our analysis here. |
| The validation exercise by comparing with Carbon Tracer is a bit strange, because the LPJ simulated biosphere fluxes will already give a reasonable correlation with CT or any inversion for that matter. | |

| **Comment 5:** | See our response to Comment 2. We have removed many statements and (at times) full sections that we deemed were more descriptive than what was needed. |
| The manuscript draft should be revised in a less descriptive way in my opinion before consideration for publication, e.g., use more Tabular contents even for the experimental description. | |